# Spectrum of Elementary Cellular Automata and Closed Chains of Contours †

**Alexander Tatashev** [1,2,‡,§] **and Marina Yashina** [1,2,*,‡,§] 

1   Department of Higher Mathematics, Moscow Automobile and Road Construction State Technical University (MADI), Moscow 125319, Russia; mv.yashina@madi.ru or a-tatashev@yandex.ru
2   Department of Mathematical Cybernetics and IT, Moscow Technical University of Comminications and Informatics (MTUCI), Moscow 111024, Russia
*   Correspondence: yash-marina@yandex.ru or yamv@mtuci.ru
†   This paper is an extended version of our paper published in Buslaev, A.P.; Tatashev, A.G.; Fomina, M.J.; Yashina, M.V. On Spectra of Wolfram Cellular Automata in Hamming Spaces. In Proceedings of the 6th International Conference on Control, Mechatronics and Automation, Tokyo, Japan, 12–14 October 2018; pp. 123–127.
‡   Current address: Department of Higher Mathematics, Moscow Automobile and Road Construction State Technical University (MADI), 64, Leningradky pr., Moscow 125319, Russia.
§   These authors contributed equally to this work.

**Abstract:** In this paper, we study the properties of some elementary automata. We have obtained the characteristics of these cellular automata. The concept of the spectrum for a more general class than the class of elementary automata is introduced. We introduce and study discrete dynamical systems which represents the transport of mass on closed chains of contours. Particles on contours move in accordance with given rules. These dynamical systems can be interpreted as cellular automata. Contributions towards this study are as follows. The characteristics of some elementary cellular automata have been obtained. A theorem about the velocity of particles' movement on the closed chain has been proved. It has been proved that, for any $\varepsilon > 0$, there exists a chain with flow density $\rho < \varepsilon$ such that the average flow particle velocity is less than the velocity of free movement. An interpretation of this system as a transport model is given. The spectrum of a binary closed chain with some conflict resolution rule is studied.

**Keywords:** cellular automata; spectrum of dynamical system; mass transport dynamical systems

## 1. Introduction

*Cellular automaton* is a dynamical system with discrete state space and discrete time scale. Cellular automaton contains *cells,* and, at any time, each cell is in one of the states belonging to a finite set. For example, each cell of a binary cellular automaton can be in one of two states: 0 or 1. For each cell, a set of neighboring cells is defined, and this set is called the *neighborhood* of the cell. The state of a cell of the cellular automaton at time $t + 1$ depends on states of cells at time $t$ such that these cells belong to the neighborhood.

J. von Neumann and S. Ulam introduced cellular automata as models of biological systems [1–3]. Cellular automata have applications in physical, biological and computational systems. The concept of elementary cellular automaton was introduced by S. Wolfram [4].

In [5], the properties of elementary automata were studied. The concept of the spectrum of elementary automata was introduced in [6]. The spectrum of some elementary cellular automata was studied in [6].

A dynamical system called a binary closed chain of contours was introduced and studied in [7]. This dynamical system was considered in [8] as an example of cellular automaton. It has been noted in [8] that this system is the elementary cellular automaton CA 063 (or ECA rule 063). In [8], the spectrum of other elementary cellular automata was studied.

In this paper, we continue to study the spectrum of elementary automata and introduce the spectrum of a class of cellular automata. We study a generalization of the binary closed chain of contours. This system belongs to the class of generalized elementary automata. We also study a version of a binary closed chain of contours, which is the elementary cellular automaton CA 029.

In Sections 2 and 3, the concept of the spectrum of an elementary cellular automaton is introduced. In Section 4, we give a concise overview of traffic models based on cellular automata.

In Section 5, we describe a dynamical system called a binary chain of contours. This system is called a closed chain of contours with the left–priority conflict resolution rule. This system is equivalent to CA 063. The concept of the average velocity of particles is introduced. We have proved that the value of the average velocity and the eigenvalue of CA 063 are the same. In Section 6, we prove theorems regarding the properties of some elementary automata. In particular, we study the spectrum of elementary cellular automata.

In Section 7, we study the properties of a version of a binary closed chain of contours. This system is equivalent to CA 029. A theorem regarding the the spectrum of the system has been proved.

In Sections 8 and 9, we introduce the concept of the spectrum that is defined for a more general class of cellular automata more general than the class of elementary cellular automata. This system is interpreted as a cellular automaton. We prove a theorem regarding the properties of the system. Let us describe a transport interpretation of this system. Suppose that raw materials or fuel is delivered to subdivisions of a factory from warehouses by vehicles, for example trucks moving on factory railways. Each vehicle moves on its line. The lines cross at points such that warehouses are located in these points. If one of the vehicles comes to a warehouse while another vehicle is being loaded, the former vehicle is waiting for loading to be completed and then is loaded itself. Then, the vehicles' movement is modeled by a dynamical system of the class to this system.

## 2. Concept of Elementary Cellular Automaton

Assume that there is a one-dimensional infinite or circular sequence of cells. At any time $t = 0, 1, 2, \ldots$, each cell is in the state 0 or 1. The state of the cell $i$ at time $t + 1$ depends only on the states of cells $i$ and $i + 1$ at time $t$. Elementary cellular automaton is determined by the following table

$$111 \to a_7, \ 110 \to a_6, \ 101 \to a_5, \ 100 \to a_4,$$

$$011 \to a_3, \ 010 \to a_2, \ 001 \to a_1, \ 000 \to a_0.$$

The number $N$ of cellular automaton is calculated by the formula

$$N = \sum_{i=0}^{7} a_i \cdot 2^i.$$

There are 256 elementary cellular automata $CA\ 000, CA\ 001, \ldots, CA\ 255$.

We consider a set of $n$-dimensional vectors $x = (x_1, \ldots, x_n)$ with binary coordinates $x_i \in \{0, 1\}$. We introduce the distance function

$$\rho(x, y) = \frac{1}{n} \sum_{i=1}^{n} |x_i - y_i|. \tag{1}$$

### 3. Spectrum Wolfram Elementary Cellular Automata

Suppose that a cellular automaton is defined on a closed circle. Since the cellular automaton is a deterministic system with a finite state space, the cellular automaton states are repeated periodically from some moment. A periodic trajectory in the state space of the cellular automaton is called a *spectral cycle*. The spectral cycle depends on the initial state $x(0)$. Let us denote by $T = T(x(0))$ the period of the spectral cycle. Suppose $\Delta x(t) = \rho(x(t), y(t))$, where $\rho$ is defined in accordance with (1); then the value

$$\frac{1}{T} \sum_{t=t_0}^{T-1} \Delta x(t) = \lambda(x(0))$$

is the *eigenvalue* of the spectral cycle.

We will give the following definition. The number $\lambda(x(0))$ is called the *eigenvalue of the spectral cycle*. The pair $< \lambda, X(x(0)) >$ is called a *spectral pair*. The set of spectral pairs for different initial states is called the *spectrum* of the elementary cellular automaton. The set of eigenvalues for different initial states is called the *spectrum* of the cellular automaton eigenvalues.

### 4. Traffic Models and Cellular Automata

The well-known Nagel-Schreckenberg transport model, which was introduced in [9], is a cellular automaton. Analytic results for a simple version of the Nagel–Schreckenberg model have been obtained in [10]. It is noted in [10] that the model is equivalent to the elementary cellular automaton rule 184 (CA 184, ECA rule 184) in the classification of Wolfram [4].

We shall obtain the spectrum of eigenvalues of the cellular automaton $CA$ 184 in Section 6.

Results, similar to results of [10], have been obtained independently in [11]. In accordance with results of [10,11], all particles move after some moment at every time for any initial state if the density of particles (the number of particles divided by the number of cells) is not more than $1/2$. The average velocity of particles (the average number of transitions of a particle per time unit) equals $(1 - \rho)/\rho$, where $\rho$ is the density. In [12], analytical results have been obtained for more general traffic model. In this model, a particle moves from the cell $i$ to the vacant cell $i + 1$ ahead of particle with probability depending on states of the cells $i - 1$, $i + 2$ (cells are numbered in the direction of movement). In [12], the behavior of particles has been studied for some particular cases, and, in the general case, the formula for velocity has not been obtained. In [13] (Kanai, 2008), a formula has been obtained for a stochastic version of the traffic model. In this system, at every step, each particle moves onto a cell forward if the cell ahead is vacant.

A two-dimensional traffic model with a toroidal supporter (BML traffic model) has been introduced in [14] (Biham, Middelton, Levin, 1992). In this model, particle move in accordance with a rule, similar to the rule CA 184. Conditions of self-organization (every particle moves after some moment) and collapse (no particle moves after some moment) for the BML model were obtained in [15,16].

In [17], a graph cellular automaton with a variable configuration of cells was introduced. The developed mechanism enables modelling of phenomena found in complex systems (e.g., transport networks, urban logistics, social networks) taking into account the interaction between the existing objects.

In [18,19], a dynamical system was studied. There are two contours with common point (node). Particles of each contour are located in cells. The particles move in accordance with the rules CA 184 or CA 240. Particles cannot cross the node simultaneously. Therefore delays occur at the node.

### 5. Binary Closed Chains of Contours and CA 063

In [7], *a closed chain of contours* has been studied, Figure 1. Each contour of this system has common points (*nodes*) with two neighboring contours. There are two cells and a particle on each contour.

In every discrete time moment, the particle of each contour is in the upper cell (cell 1) or the lower cell (cell 0).

Particles cannot cross a common node simultaneously. A competition of particles occurs if the particle of the contour located to the right of the contour, tends to go from the upper cell to the lower cell, and the particle to the left of the node tends to go from the lower cell to the upper cell. In [7], different rules of competition resolution have been considered. One of them is the right-priority rule. In accordance with the rule, the particle located only to the right of the common node moves, and the left contour particle waits. The system state at each time is a cyclic vector of zeros and ones, where the $i$th coordinate takes the value 0 or 1 depending on the number of the cell occupied by a particle of the $i$th contour. Such dynamical system is equivalent to the cellular automaton CA 119 on a closed sequence of $n$ cells. If in the case of a competition, the particle located to the left contour of the common node moves (the left-priority rule), then the dynamical system is equivalent to CA 063.

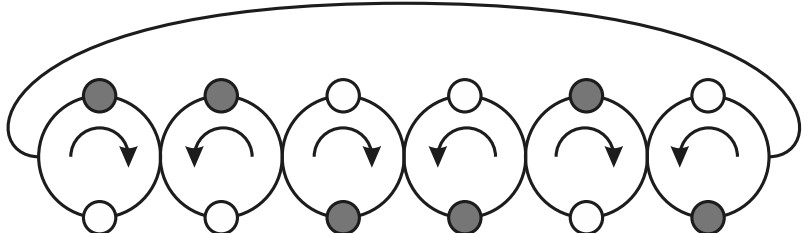

**Figure 1.** A closed chain of contours, CA 063, state 110010.

If we replace the counter-clockwise direction of movement to the movement with clockwise direction, then dynamical system is equivalent to CA 017 (the right-priority rule) or CA 003 (the left-priority rule). Since the dynamical system is deterministic with finite set of states, there exists some cyclic trajectory in the state space of the system, Figure 2. The portion of particles moving on the cycle, averaged over the states on the cyclic trajectory, is called average velocity of particles' movement. In general case the limit cyclic trajectory and the average movement velocity depend on the initial state of the system. A set of possible values of the average velocity of movement for different initial states is called the *spectrum of velocities.*

Let us denote by $[a]$ the integral part of number $a$. In accordance with results obtained in [7] for closed chains of contours with the rules CA 003, CA 017, CA 063, CA 119, the spectrum of velocities contains $[n/3] + 1$ values. These values are

$$v = 1 - \frac{k}{n}, \; k = 0, 1, \ldots, \left[\frac{n}{3}\right],$$

where $n$ is the number of contours. The spectrum of eigenvalues for the cellular automata CA 003, CA 017, CA 063, CA 119 contains $[n/3] + 1$ values

$$\lambda = v = 1 - \frac{k}{n}, \; k = 0, 1, \ldots, \left[\frac{n}{3}\right].$$

**Theorem 1.** *Suppose the number of a binary closed chain contours is n, and the initial state is a vector*

$$(\alpha_1(t), \ldots, \alpha_n(t)),$$

*where, if the particle of the ith contour is in the lower cell at time t, then $\alpha_i(t) = 0$, and, if the particle of the i-th contour is in the upper cell, then $\alpha_i(t) = 1$. Particles move counter-clockwise, and the conflict resolution rule is the right-priority rule. Then the average velocity of particles is equal to the eigenvalue of the relative spectral cycle of CA 063 that is defined on the ring, containing i particles.*

**Proof.** Suppose the initial states of the closed chain and the cellular automaton are $(\alpha_1(0), \ldots, \alpha_n(0))$, $(x_1(0), \ldots, x_n(0))$ respectively, and $\alpha_i(0) = x_i(0)$. Then we have $\alpha_i(t) = x_i(t)$, $i = 1, \ldots, n$ for any $t = 0, 1, 2, \ldots$ The coordinate $x_i(t)$ is changed at time $t$ if and only if the particle of the $i$-th contour moves at time $t$. From this, the theorem follows.   □

A system of similar type was studied in [19]. In this system there are $m$ cells and a cluster containing $k < m$ particles on each contour. A continual analogue of the system, considered in [20], has been studied in [21].

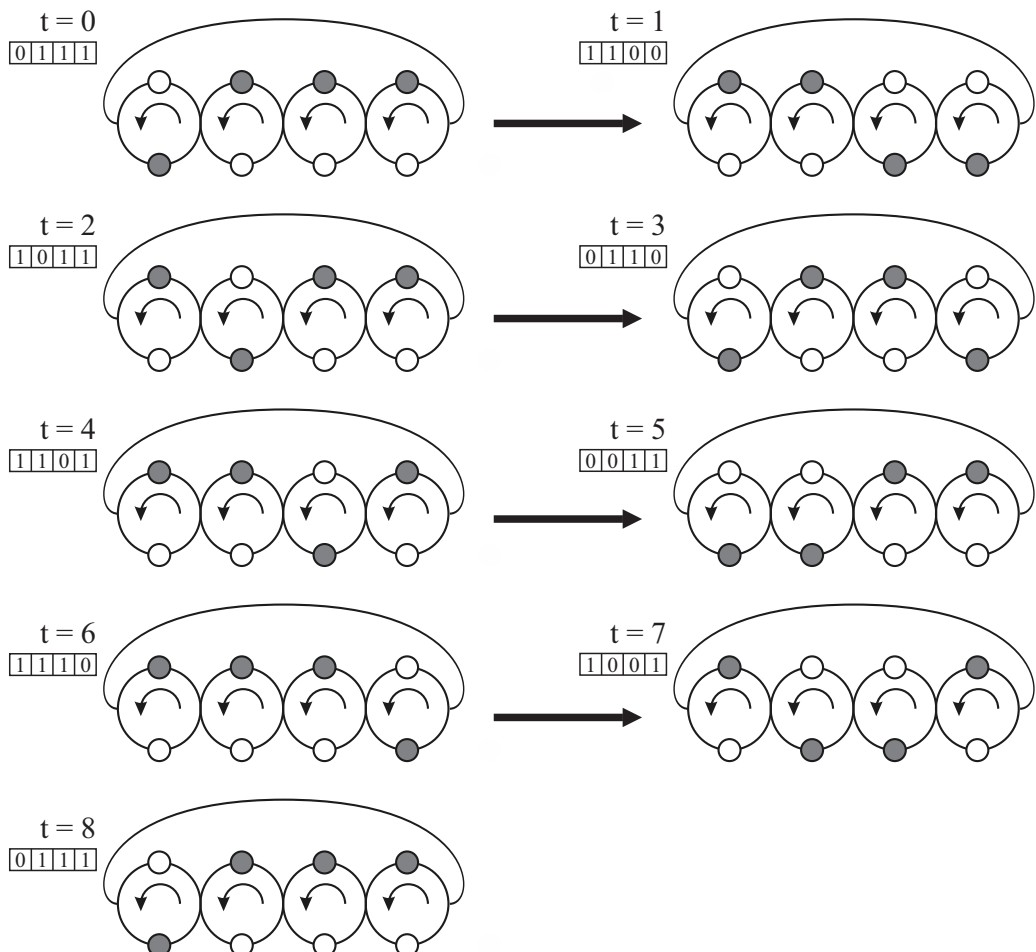

**Figure 2.** A spectral cycle for a closed chain of contours with rule CA 063.

## 6. Spectrum of Some Elementary Automata

We have obtained the following results for elementary cellular automata. We assume that each cellular automaton is defined on a closed chain containing $n$ cells.

**Theorem 2.** *For cellular automata CA 002, CA 016, CA 191, CA 247, the following is true. Every cyclic state vectors on each spectral cycle do not contain clusters of units "1" (for cellular automata CA 002, CA 016) or clusters of zeros "0" for cellular automata CA 191, CA 247 with length more than 1, and, in the cyclic vector, between each two units (zeros) there are at least two zeros (ones), and the number of ones (zeros) in the vector does not exceed $[n/3]$. Each spectral cycle, satisfying this condition, belongs to some spectral cycle. There are $[n/3] + 1$ eigenvalues. These eigenvalues are*

$$1 - \frac{k}{n}, \ k = 0, 1, \ldots, [n/3].$$

*On each spectral cycle, the cyclic state vector shifts one position to the left (for cellular automata CA 002, CA 191) or to the right (for cellular automata CA 002, CA 191).*

**Proof.** Consider the rule CA 002

$$111 \rightarrow 0, \ 110 \rightarrow 0, \ 101 \rightarrow 0, \ 100 \rightarrow 0,$$

$$011 \rightarrow 0, \ 010 \rightarrow 0, \ 001 \rightarrow 1, \ 000 \rightarrow 0.$$

If at time $t + 1$ the cell $i$ is in the state 1 ($x_i(t + 1) = 1$), then $x_{i-1}(t) = x_i(t) = 0$, $x_{i+1}(t) = 1$ (indexes are calculated by modulo $n$). Therefore, in accordance with rule CA 002, $x_{i-2}(t + 1) = x_{i-1}(t + 1) = x_{i+1}(t + 1) = x_{i+2}(t + 1) = 0$. From these equalities, Theorem 2 follows for CA 002. In the cases of the rules CA 016, CA 191, CA 247, the theorem is proved similarly. $\square$

Theorem 3 is related to the cellular automata that can be interpreted as the closed chains of contours, described in Section 2.

**Theorem 3.** *For the cellular automata CA 003, CA 017, CA 063, CA 119, the following is true. Every cyclic state vector belongs to a spectral cycle if and only if this cycle does not contain clusters of ones "1" (for the cellular automata CA 063, CA 017) or clusters of zeros "0" for cellular automata CA 003, CA 119 of length less than 2. There are $[n/3] + 1$ eigenvalues. These eigenvalues are*

$$1 - \frac{k}{n}, \ k = 0, 1, \ldots, [n/3].$$

*On each spectral cycle, the cyclic state vector shifts one position to the left (for cellular automata CA 017, CA 119) or to the right (for cellular automata CA 003, CA 063) for two steps.*

Theorem 3 follows from results of [7].

**Theorem 4.** *There are $[n/2] + 1$ eigenvalues of cellular automata CA 170, CA 240. These eigenvalues are $2k/n, \ k = 0, 1, \ldots, [n/2].$*

**Proof.** State cyclic vector of cellular automata CA 170, CA 240 shifts onto one position to the left at each step (for CA 170) or to the right (for CA 240). Suppose $k$ is the number of "1" clusters in the state cyclic vector. At any step, just two cells change their states. There can be not more than $[n/2]$ contours. Therefore the eigenvalues of the cellular automaton spectrum are equal to $2k/n, \ k = 0, 1, \ldots, [n/2]$, and therefore the number of eigenvalues equals $[n/2] + 1$. Theorem 4 has been proved. $\square$

**Theorem 5.** *There are $[n/2] + 1$ eigenvalues of cellular automata CA 015, CA 085. These eigenvalues are $1 - 2k/n, \ k = 0, 1, \ldots, [n/2], \ k = 0, 1, \ldots, [n/2].$*

**Proof.** For cellular automata $CA$ 015, $CA$ 085, at time $t + 1$, the state of the cell $i$ is opposite to the state of the cell $i - 1$ (for CA 015) or the state of the cell $i$ is opposite to the state of the cell $i + 1$ (for CA 085). Therefore, in accordance with rule CA 015 (rule CA 085), any cell changes its state if and only if this cell does not change its state in the case of rule CA 240 (rule 170). From this and Theorem 4, Theorem 5 follows. $\square$

**Theorem 6.** *There are $[n/2] + 1$ eigenvalues of the cellular automata CA 034, CA 048, CA 187, CA 243. These eigenvalues are $2k/n, \ k = 0, 1, \ldots, [n/2], \ k = 0, 1, \ldots, [n/2]$. On any spectral cycle of each of these cellular automata, the cyclic vector any state cannot contain clusters of "1" (for CA 034, CA 048) or clusters of "0" (for CA 187, CA 243) of length more than 1, and the cyclic vector shifts onto one position to the left (for CA 034, CA 187) or the cyclic vector shifts to one position to the right (for CA 048, CA 243).*

**Proof.** For rules CA 034, CA 048, CA 187, CA 243, the cellular automaton results in a state such that the cyclic vector of this state does not contain clusters of "1" (for CA 034, CA 048) or clusters of "0" (for CA 187, CA 243) of length more than 1. After this the cyclic vector of the system state shifts onto one position at every step. There are no more than $k$ clusters. If $k$ is the number of "1" clusters in the state cyclic vector, then the eigenvalues of the cellular automaton spectrum is equal to $2k/n$, $k = 0, 1, \ldots, [n/2]$, and therefore the number of eigenvalues equals $[n/2] + 1$. Theorem 6 has been proved. □

**Theorem 7.** *The unique eigenvalue of the spectrum of the cellular automata CA 004, CA 012, CA 068, CA 207, CA 221, CA 223 is equal 0, and, over no more than one step, each of these cellular automata results in a state such that there are no clusters of "1" (for CA 004, CA 012, CA 068) or clusters of "0" (for CA 207, CA 221, CA 223) of length more than 1. After this, the system state does not change (the length of each spectral cycle equals 1).*

**Proof.** For the rule CA 004, Theorem 7 follows from that the cell $i$ is in the state 1 at time $t + 1$ if and only if

$$x_i(t) = 1, \; x_{i-1}(t) = x_{i+1}(t) = 0.$$

For the rule CA 223, Theorem 7 is proved similarly. □

For the rule CA 012, at each moment the length of every cluster of "1" with length more than 1 is reduced by 1 while there is at least one cluster of "1" with length more than 1. If the system is in the state such that the cyclic vector of this state contains no cluster of "1" with length more than 1, then the system state does not change. From this, Theorem 7 follows for the rule CA 012. For the rules CA 068, CA 207, CA 221, Theorem 7 is proved similarly.

**Theorem 8.** *The unique eigenvalue of the spectrum of cellular automata CA 072, CA 237 is equal 0, and, over no more than one step, each of these cellular automata results in a state such that there are no cluster of "1" with the length not equal to 2 (for CA 072) or clusters of "0" (for CA 237) of length not equal to 2. After this, the system state does not change (the length of each spectral cycle equals 1).*

**Proof.** For the rule CA 072, Theorem 8 follows from that over one step the system results in the state such that the state vector does not contain any cluster of "1" of length not equal to 2. After this, the system state does not change. For the rule CA 237, Theorem 8 is proved similarly. □

**Theorem 9.** *The unique eigenvalue of the spectrum of cellular automata CA 076, CA 205 is equal 0, and, over no more than one step, each of these cellular automata results in a state such that there are no cluster of "1" with the length less than 3 (for CA 072) or clusters of "0" (for CA 205) of length of 1. After this, the system state does not change, i.e. the length of each spectral cycle equals 1.*

Theorem 9 is proved similarly to Theorem 8.

**Theorem 10.** *The unique eigenvalue of the spectrum of cellular automata CA 008, CA 064, CA 239, CA 253 is equal to 0, and each of these cellular automata results in the state $(0, 0, \ldots, 0)$ (for CA 008, CA 064) or $(1, 1, \ldots, 1)$ (for CA 239, CA 253) for a finite time.*

**Proof.** The length of the clusters of "1" (for CA 008, CA 064) or "0" (for CA 239, CA 253) in the cyclic vector is reduced by 1 at any step. From this, Theorem 10 follows. □

**Theorem 11.** *Cyclic vectors of states on each spectral cycle of the cellular automata CA 010, CA 014, CA 024, CA 066, CA 080, CA 084 do not contain clusters of "1" of length 2. Cyclic vectors of states on each spectral*

*cycle of cellular automata CA 143, CA 175, CA 189, CA 213, CA 231, CA 245 do not contain clusters of "0" of length 2.*

**Proof.** Consider the rule CA 010

$$111 \rightarrow 0, \ 110 \rightarrow 0, \ 101 \rightarrow 0, \ 100 \rightarrow 0,$$

$$011 \rightarrow 1, \ 010 \rightarrow 0, \ 001 \rightarrow 1, \ 000 \rightarrow 0.$$

In accordance with this rule, over one step, the cellular automaton results in a state such that the cyclic vector of this state contains no cluster of length more than 2. From this, Theorem 10 follows in the case of the rule CA 014. In the case of the rules CA 024, CA 066, CA 080, CA 084, CA 143, CA 175, CA 189, CA 213, CA 231, CA 245, Theorem 11 is proved similarly. □

**Theorem 12.** *Spectral cycles of cellular automaton CA 184 and CA 226 can have only following eigenvalues*

$$\frac{2k}{n}, \ k = 0, 1, \ldots, \left[\frac{n}{2}\right].$$

**Proof.** The cellular CA 184 can be interpreted as a traffic model, which has been considered in [7]. The eigenvalue of each spectral cycle depends only on the number of "1" in the cyclic vector of each state on the spectral cycle (the number of particles in the traffic models). This eigenvalue equals the average velocity of the particle multiplied by $\frac{2m}{n}$. For the rule 184, Theorem 12 follows from this and results of [7,8] (Section 4, Equation (1)). Since CA 226 is equivalent to the traffic model such that in this model is similar to the traffic model with rule CA 184 but particles move in the opposite direction. □

**Theorem 13.** *For any spectral cycle, the only eigenvalue of the spectrum of cellular automata CA 051 is equal to 1 and the period of the spectral cycle equals 1.*

**Proof.** Theorem 13 follows from that the cellular automaton CA 051 changes at each step. □

**Theorem 14.** *The only eigenvalue of the spectrum of cellular automata CA 000 or CA 255 is equal 0.*

**Proof.** Theorem 14 follows from that the state vector, except the vector of the initial state, cannot contain "1" in the case of CA 000 ("0" in the case of CA 255). □

## 7. A Version of Binary Closed Chain, CA 029

Let us consider a version of the binary closed chain of contours. In Section 5, we considered a binary closed chain such that, if two particles tend to cross the same node, then only one of the particles moves. In Section 7, we assume that, if two particles tend to cross the node simultaneously, then none of these particles move at the current moment and in the future. Assume that, if the particle of the contour is in the lower cell, then the contour is in the state 0, and, if the particle of is in the upper cell, then the contour is in the state 1. Let each contour correspond to a cell of a cellular automaton. If each particle moves counter-clockwise, then the system is equivalent to the elementary cellular automaton CA 029, and, if particles move clockwise, then the system is equivalent to the elementary cellular CA 071. Let the number of contours (cells of the cellular automaton) be equal to $n$.

**Theorem 15.** *The spectrum of velocities of the cellular automaton CA 029 or CA 071 contains values $1 - \frac{2k}{n}$, $k = 0, 1, \ldots, \left[\frac{n}{2}\right]$. The system over one step results in the state belonging to the spectral cycle. On each spectral cycle, each particle does not move at any moment or moves at any moment.*

**Proof.** We consider the cellular automaton CA 029. In the case of CA 071, the proof is the same. Let us consider the corresponding binary chain of contours. Suppose, in the initial state, there are $k_1$

neighboring pairs of non-moving particles. For each of these pairs, the left contour is in the state 0, and the right contour is in the state 1. Suppose, in the initial case, there are $k_2$ neighboring pairs of moving particles. For each of these pairs, the left contour is in the state 1, and the right contour is in the state 0. At the next moment, the system is in the state such that there are $k_1 + k_2$ pairs of non-moving particles, and the other particles move at the current moment and any moment in the future. From this, Theorem 15 follows.　□

## 8. Concept for Spectrum for Generalization of Elementary Cellular Automaton

In [4], the following generalization of elementary cellular automata has been introduced. Each cell of a cellular automaton is at any time in one of $k$ states. The state of the cell $i$ at time $t + 1$ depends on the states of the cells $i - 1$, $i + 1$ at time $t$. The state of such automaton is a vector $x = (x_1, x_2, \ldots, x_n)$, where $x_i \in \{1, 2, \ldots, k\}$.

We shall give the following definitions. Suppose $x = (x_1, x_2, \ldots, x_n)$, $y = (y_1, y_2, \ldots, y_n)$,

$$\rho(x, y) = \frac{1}{n} \sum_{i=1}^{n} |x_i - y_i|.$$

The average value of the $\rho(x, y)$ is called *an eigenvalue.*

A set of periodically repeated states is called a *spectral cycle.* The *spectrum* is the set of spectral cycles and related eigenvalues.

In Section 9, a generalization of binary closed chains (Section 6) will be considered.

## 9. Generalization of Binary Closed Chain

We consider a discrete dynamical system, containing $n$ *contours* $0, 1, \ldots, n - 1$. There are $2k$ *cells* $0, 1, \ldots, 2m - 1$ and one *particle* on each contour, Figure 3. Cells are numbered in the direction of the particle movement. There is a common point (*node*) of the contours $i$ and $i + 1$ (addition by modulo $n$), $i = 0, 1, \ldots, n - 1$. The common node of contours $i$ and $i + 1$ is located between the cells 0 and 1 on the contour $i$ and between the cells $m$ and $m + 1$ on the contour $i + 1$. Two particles cannot cross the common node simultaneously. At any moment $t = 0, 1, 2, \ldots$, each particle moves onto one position in the direction of movement if no delay occurs. If the particle of the contour $i$ (*particle $i$*) is in the cell 0 and the particle $i + 1$ is in the cell $m$, then *a conflict* of particles $i$ and $i + 1$ occurs. In the case of the conflict, the particle $i$ moves and the particle $i + 1$ does not move. Therefore, if, at time $t$, the particle $i$ is in the cell 0 and the particle $i + 1$ is in the cell $m$, then, at the time $t + 1$, the particle $i$ will be at the cell 1, and the particle $i + 1$ will be at the cell $m$. The initial state of the system is given. The value $\rho = \frac{1}{2m}$ is called the *density* of particles flow. A state of the system at time $t$ is the vector

$$(\alpha_0(t), \ldots, \alpha_{n-1}(t)),$$

where $\alpha_i$ is the index of the cell in that the particle $i$ is located at time $t$. The initial state of the system is given.

The average number of transitions of a particle per time unit is called *the average velocity* of the particle.

We say that the system is at time $t_0$ at a *state of free movement* if at any time $t \geq t_0$ all particles move. If the system results in a state of free movement, then the velocity of particles is equal to 1. If the system results in a state of free movement, then the average velocity of particles is less than 1.

This dynamical system is a cellular automaton of the type considered in Section 7. The average velocity of particles is equal to the eigenvalue of the cellular automaton for the relative initial state.

**Theorem 16.** *For any $\varepsilon > 0$, there exists a number $2m$ (number of cells in any contour) and a number $n$ (number of contours) and an initial state such that*

$$\rho = \frac{1}{2m} < \varepsilon,$$

*and the average velocity of particles $v < 1$.*

**Proof.** Suppose the number of cells in each contour is equal to $2m = 4s$, the number of contours is equal to $2s + 1$, where $s$ is a natural number.

The initial state is

$$\alpha_0(0) = 0, \ \alpha_1(0) = m,$$

$$\alpha_i(0) = m + (i - 1)(m - 1), \ i = 2, 3, \ldots, n - 1,$$

(addition by modulo $2m$). At time $t = 0$, the particle 1 does not move, and the other particles move. Hence,

$$\alpha_0(1) = 1, \ \alpha_1(1) = m,$$

$$\alpha_i(1) = m + 1 + (i - 1)(m - 1), \ i = 2, 3, \ldots, n - 1,$$

(addition by modulo $2m$). At time $t = 1, \ldots, m + 1$, all particles move, and, at the time $t = m + 1$, the system is in the state

$$\alpha_0(m + 1) = m + 1, \ \alpha_1(m + 1) = 0, \ \alpha_2(m + 1) = m,$$

$$\alpha_i(m + 1) = m + (i - 2)(m - 1), \ i = 3, 4, \ldots, n - 1.$$

We have

$$\alpha_{n-1}(0) = m + (n - 2)(m - 1) = m + (n - 2)m - n + 2 =$$

$$= (n - 3)m + 2m - n + 2 = 4(s - 1)s + 2s + 1 = 4(s - 1)s + m + 1,$$

and therefore

$$\alpha_{n-1}(0) = \alpha_0(m + 1) \pmod{2m}.$$

Thus we obtain the vector of state at time $t$ by the cyclic shift of the vector of the initial state onto a position to the right, Figure 3.

At time $t = (m + 1)^2 = n^2$, the system returns to the initial state

$$\alpha_0\left(n^2\right) = 0, \ \alpha_1\left(n^2\right) = m,$$

$$\alpha_i\left(n^2\right) = m + (i - 1)(m - 1), \ i = 2, 3, \ldots, n - 1.$$

At time segment $[0, n^2 - 1]$, one delay of each particle occurs. The average velocity of particles equals

$$v = 1 - \frac{1}{n^2}.$$

The density is

$$\rho = \frac{1}{4s}.$$

If

$$s > \frac{1}{4\varepsilon},$$

then $\rho < \varepsilon$. Theorem 16 has been proved. $\square$

Thus, in accordance with Theorem 16, *for any small density of particles, there exists n such that from some initial states the system does not result in a state of free movement over a finite time.*

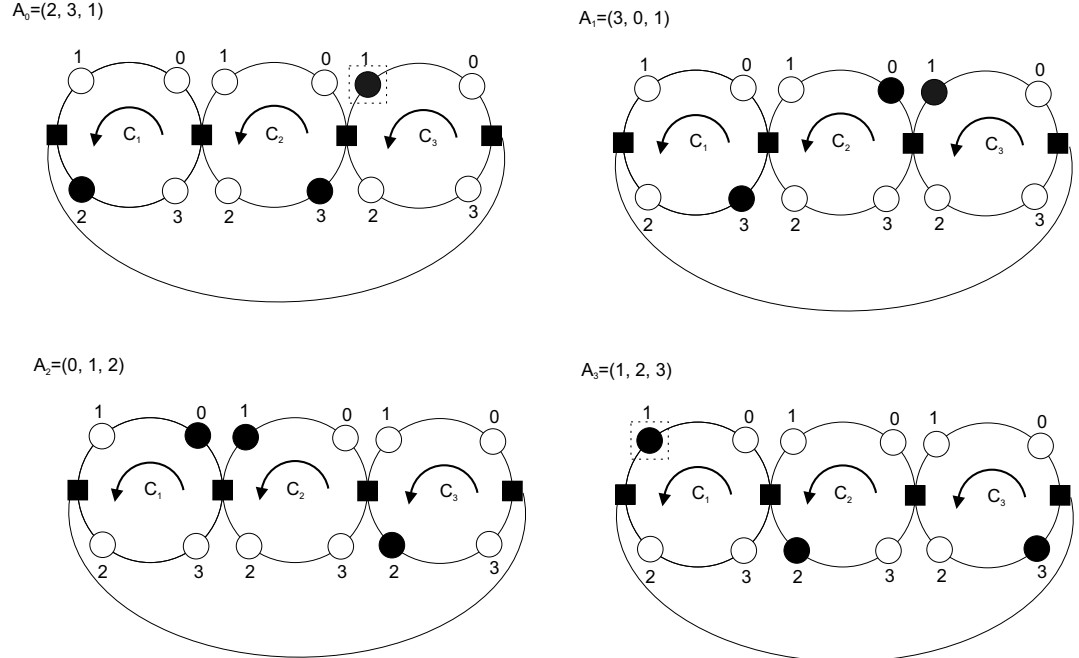

**Figure 3.** A spectral cycle for a closed chain of contours with 4 cells on each contour.

## 10. Conclusions and Future Works

We have studied the spectrum of some elementary cellular automata. We introduce and study a dynamical system, which is a version of a binary chain of contours. We have defined the concept of the spectrum for a more general class of cellular automata than the class of elementary cellular automata. We study a generalization of the binary closed chain of contours. This generalized binary closed chain belongs to the class of cellular automata, considered in Section 8.

We plan to continue to study the spectrum of elementary cellular automata. We also plan to study the dynamics of cellular automata CA 136 and CA 252. As it noted in [22], the cellular automaton CA 252 models a section at intersections before a red light, and the cellular automaton CA 136 models a section after a red light. We also plan to study stochastic versions of cellular automata. If, in the deterministic version of a cellular automaton, the current state of the cellular automaton is chained, then, in the stochastic version, the state is changing with probability $p$, $0 < p < 1$. A stochastic version of the binary closed chain of contours, equivalent to the cellular automaton CA 063, has been introduced and studied in [23].

**Author Contributions:** Conceptualization, A.T. and M.Y.; methodology, A.T. and M.Y.; validation, A.T. and M.Y.; formal analysis, A.T. and M.Y.; writing—original draft preparation, A.T. and M.Y.

**Funding:** This research was funded by of the Russian Foundation for Basic Research grant number 17-01-00821-a and number 17-07-01358-a.

**Conflicts of Interest:** The authors declare no conflict of interest.

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
