# Peer review of "Spectrum of Elementary Cellular Automata and Closed Chains of Contours"

_machines, doi:10.3390/machines7020028_

Round 1

Reviewer 1 Report

The introduction lacks brief explanations of the concepts presented in the paper. Even though they are given further into the paper, it would be interesting for the reader to be able to have a basic understanding of the concepts so that the results and its implications can be more readily understood without having to delve into the work itself. In this case, giving a brief overview of what elementary cellular automata and their spectrum are, and briefly explaining the binary closed chain of contours and its uses, would fulfill this recommendation.

The abstract is missing key information. Does this "more general class of elementary cellular automata" have a name? If so, it should be used. What is the purpose of the presented dynamical system? It appears it is being used to model traffic, so make that clear. What is the implication of the average flow velocity being less than the free movement velocity for the systems presented in the paper? Include a brief explanation of the results and their relevance.

The paper presents a compilation of theorems and their proofs, however, some of the proofs lack deeper explanations. A few of them are obvious enough to be easily comprehended, but others need more development. Moreover, the paper does not present discussions of the results (the theorems and their proofs), so the reader is left guessing what those theorems are being used for and what they imply. It appears they are being used for modeling traffic intersections (this can only be inferred from the references), however, the paper does not explain the consequences of the theorems when applied to the problem.

The conclusions presented are simplistic and revolve around "concepts were presented, theorems were demonstrated and a model was studied", and again, the paper does not present any discussion over the results and their relevance. The authors should discuss the results and their implications somewhere in the paper, otherwise, the reader is left guessing the purpose of this work.

The paper is readable, however, there are missing connections between a lot of words, especially in the theorems and proofs. Moreover, some phrases are confusing (i.e. on page 4, "Let us denote by [a] the integer part of [a]", and on page 5, "the cyclic vector of this state does not contain clusters of '1' or clusters of '1'"). The authors should review the text extensively, especially the theorems, to fix these errors.

Regarding the style, the authors use indirect speech abundantly making the paper monotonous and impairing the reader's comprehension. The text should be revised seeking to use direct speech wherever possible.

Author Response

Thank you so much for deep  comments.

We have made corrections in the text of the paper

and, in particular, in the text of proofs. We have extended the

introduction and the list of references.

{\bf P.2. Section 1.} In this paper, we continue to study the spectrum

of elementary automata and introduce the spectrum of a class  of cellular automata. We

study a generalization of the binary closed chain of contours. This system belongs to

the class of generalized elementary automata.

We also study a version of a binary closed chain

of contours, which is the elementary cellular automaton CA~029.

In Sections 2, 3, the concept of the spectrum of an elementary cellular

automaton is introduced.

In Section 4, we give a concise overview on traffic models based on cellular

automata.

In Section 5, we describe a dynamical system called a binary chain of

contours. This system is called a closed chain of contours with the left--priority

conflict resolution rule. This system is equivalent to CA~063. The concept of the average

velocity of particles is introduced. We have proved that the value of the average

velocity and the eigenvalue of CA~063 are the same.

In Section~6, we prove theorems about properties of some elementary

automata. In particular, we study the spectrum of elementary cellular

automata.

In Section 7, we study properties of a version of binary closed chain of contours.

This system is equivalent to CA~029. A theorem about the spectrum of the system

has been proved.

In Sections 8, 9, we introduce the concept of the spectrum is defined for a class of cellular

automata more general than the class of elementary cellular automata.

This system is interpreted as a cellular automaton. We prove a theorem

about properties of the system. 

Let us describe a transport interpretation

of this  system. Suppose raw materials or fuel is delivered to

subdivisions of a factory from warehouses by vehicles, for example

trucks moving on factory railways. Each vehicle moves on its line.

The lines cross at points such that warehouses are located in these points.

If one of vehicles comes to a warehouse while another vehicle is being loaded,

the former vehicle is waiting for the end of loading and then is being

loaded itself. Then the vehicles movement is modeled by a dynamical system of the

class to this system.

\vskip 3pt

{\bf P.11--12.}

\section*{References}

{1] von Neumann J. 1963. The general and logical theory of automata.

In {\it J. von Neumann, Collected works}, edited by A.H. Taub, 5, 288.

\vskip 3pt

[2] von Neumann J. 1966. {\it Theory of self-reproducing automata,

edited by A.W. Burks} (University of Illions, Urnana).

\vskip 3pt

[3] Ulam S. 1974. {\it Some ideas and prospects in bio-mathematics.} Ann. Rev. Bio. 255.

\vskip 3pt

[4] Wolfram, S. {\it Statistical mechanics of cellular automata.}

Rev. Mod. Mod. Phys. 1983, {\it Vol.~55,} pp.~601--644.\\

https:/dx.doi.org.10.1003/RevModPhys.55.601

\vskip 3pt

[5] Wolfram S. Tables of cellular automaton properties.

In {\it Theory and Applications of Cellular Automata (Including Selected

papers 1983--1986)}; [Wolfram S. (Ed.)] Advanced Series on Complex

Systems 1. World Scientific Publishing, 1986, pp.~485--557.

\vskip 3pt

[6] Buslaev A.P., Tatashev A.G., Yashina M.V. {\it On cellular

automata, traffic and dynamical systems in graphs.} International

Journal of Engineering and Technology,  vol.~7, no.~2.28, pp.~351--356.\\

https://dx.doi.org/10.14419/ijet.v7i2.28.13210

\vskip 3pt

[7] Kozlov V.V., Buslaev A.P., Tatashev A.G.

{\it Monotonic walks on a necklace and coloured

dynamic vector.} International Journal of Computer

Mathematics, vol. 92, no. 9, 2015, pp. 1910~-- 1920.

1920.\\

https://dx.doi.org/1080/00207160.2014/915964

\vskip 3pt

[8] Buslaev  A.P., Tatashev  A.G., Fomina M.J.,  Yashina M.V.  On Spectra

of Wolfram Cellular Automata in Hamming Spaces. In Proceedings of the

6th International Conference on Control, Mechatronics and Automation Tokyo,

Japan — October 12 - 14, 2018, ACM New York, NY, USA, 2018. pp. 123-127.\\

doi: 10.1145/3284516.3284549

[9] Nagel K., Schreckenberg M. {\it A cellular automation models

for freeway traffic.} J. Phys. I. France 2, {\bf 1992}, pp. 2221~-- 2229.\\

https://dx.doi.org/10.1051/jp1.1992277

[10] Belitzky V., Ferrary P.A. {\it Invariant measures and convergence

properties for cellular automation 184 and related processes.}

J. Stat. Phys. (2005), vol.~118, no.~3, pp.~589--623\\

https://dx.doi.org/10.1007/s10955-044-8822-4

[11] Blank M. {\it Exact analysis of dynamical systems arising

in models of flow traffic.} Russian Math Surveys, 2000, vol.~55, no.~55,~pp.~562--563

(DOI: 10.4213/rm295) {\bf 55(5)} 562-563

[12] Gray L and Grefeath D

The ergodic theory of traffic jams. {\it J. Stat. Phys.},

(DOI: 10.1023/A:1012202706850) {\bf 105(3/4)} 413--452

[13] Kanai M, Nishinary K and Tokihiro T. Exact solution and

asymptotic behavior of the asymmetric simple exclusion

process on a ring {\it arXive.0905.2795v1 [cond-mat-stat-mech] 18 May 2009}

[14] Biham O, Middleton A A and Levine D 1992

Self-organization and a dynamical transition in traffic-flow models

{\it Phys. Rev. A. American Physical Society} (DOI 10.1103/PhysRevA.46.R6124)

{\bf 46(10)} R6124--R6127

[15] D'Souza R M (2005) Coexisting phases and lattice dependence

of a cellular automaton model for traffic flow  {\it Phys. Rev. E.

The American Physical Society} (DOI 10.1103/PhysRevA46.R6.124) {\bf 71(6)}: 066112

[16] Angel O, Horloyd A E, Martin J B. (12 August 2005).

The jammed phase of the Biham-Middleton-Levine traffic model

{\it Electronic Communication in Probability} (DOI 10.1214/ECP.v10-1148)

{\bf 10}: 167--178

[17] Malecky K. {\it Graph cellular automata with relation-based neighbourhoods

of cells for complex systems modelling: A case of traffic simulation.} Symmetry, 2017, 9,  p.~322.

https://doi.org/103390/sym912032

[18] Buslaev A.P., Tatashev A.G., Yashina M.V. About synergy of flows on flower. In Dependability

Engineering and Complex Systems. Springer, Cham {\bf 2016.} pp. 75--84.

[19] Buslaev A.P., Tatashev A.G. {\it Exact Results for discrete dynamical

system on a pair of contours.} Mathematical Methods in the

Applied Sciences, 2019, vol.~41, no.~17, pp.~7283--7204.\\

https://dx.doi.org/10.1002/mma.4822.

[20] Buslaev A.P., Fomina M.Yu., Tatashev A.G., Yashina M.V.

{\it On discrete flow networks model spectra:

statement, simulation, hypotheses.} Journal of Physics:

Conference Series, (2018), vol.~1053, 012034, pp.~1--7\\

https://dx.doi.org.10.1088/1742/6596/1053/1/012034

[21] Buslaev A.P., Tatashev A.G., Yashina M.V.

{\it Flows spectrum on closed trio of contours.} European

Journal of Pure and Applied Mathematics, 2018, vol. 11, no. 1,

pp. 260--283.\\

http://dx.doi.org.10.29020/nybg.ejpam.v11i1.3201

[22] Zubilage D., Cruz G., Aguilar L.D., Zapotecatl J., Fernandes N.,

Aguilar J., Rosenblueth D.A., Gershenson C. {\it Measuring the Complexity

of Self-Organizing Traffic Lights.} Entropy, 2014, vol.~16, no.~5, pp.~2384-2407.\\

https://doi.org/10.3390/e16052384

[23] Kozlov V.V., Buslaev A.P., Tatashev A.G. {\it A dynamical communication system

on a network.} Journal of Computational and Applied Mathematics, 2015, vol.~275, pp.~247--261.\\

https://doi.org/10.1016/j.cam.2014.07.026

Reviewer 2 Report

The article has a good scientific sound. The authors defined many theorems and proofs. Despite the good mathematical apparatus, the article is seemed to be incomplete:

What is a main contribution of the article?

What the idea is introduced for? Where it can be implemented?

There is lack of good literature review - there are 16 references and 8 of them belongs to authors. For example, section 4 describes traffic models and pointed only NaSch model and article by Belitzky and Ferrary. I have found mamy more, simple and complex, and e.g. new type of CA - Graph-based CA published in Symmetry 9(12). I think they should be mentioned, too. 

Minor weaknesses: 

ine 48. The dot is missed at the end of the sentence.

line 46, 48, 52, 57 - redundant style of reference. There should be only square brackets and the number of an article [xx].

a few commas and dots missed in the Reference section. A few references are incorrectly formatted.

Author Response

Thank you so much for deep  comments.

We have made corrections in the text of the paper

and, in particular, in the text of proofs.

We have extended the abstracts and the introduction.

We have made the following corrections.

e.

\vskip 3pt

{\bf P.1, Abstracts.} In this paper, we study the properties of some elementary automata.

We have obtained characteristics of these cellular automata.

The concept of the spectrum for more general class than

the class of elementary automata has been introduced. We introduce and study discrete dynamical

systems which are mass transport closed chains of contours. Particles on contours

in accordance with given rules. These dynamical systems can be interpreted as

cellular automata. Contributions towards this study are the following. Characteristics of some elementary cellular

automata have been obtained. A theorem about the velocity

of particles movement on the closed chain has been proved.

It has been proved that, for any $\varepsilon>0,$

there exists a chain with flow density $\rho<\varepsilon$ such

that the average flow particle velocity is less than the velocity

of free movement. An interpretation of this system as a transport

model is given. The spectrum of a binary closed chain with some

conflict resolution rule is studied.

{\bf P.2. Section.} In Sections 2, 3, the concept of the spectrum of an elementary cellular

automaton is introduced.

In Section 4, we give a concise overview on traffic models based on cellular

automata.

In Section 5, we describe a dynamical system called a binary chain of

contours. This system is called a closed chain of contours with the left--priority

conflict resolution rule. This system is equivalent to CA~063. The concept of the average

velocity of particles is introduced. We have proved that the value of the average

velocity and the eigenvalue of CA~063 are the same.

In Section~6, we prove theorems about properties of some elementary

automata. In particular, we study the spectrum of elementary cellular

automata.

In Section 7, we study properties of a version of binary closed chain of contours.

This system is equivalent to CA~029. A theorem about the spectrum of the system

has been proved.

In Sections 8, 9, we introduce the concept of the spectrum is defined for a class of cellular

automata more general than the class of elementary cellular automata.

This system is interpreted as a cellular automaton. We prove a theorem

about properties of the system. 

Let us describe a transport interpretation

of this  system. Suppose raw materials or fuel is delivered to

subdivisions of a factory from warehouses by vehicles, for example

trucks moving on factory railways. Each vehicle moves on its line.

The lines cross at points such that warehouses are located in these points.

If one of vehicles comes to a warehouse while another vehicle is being loaded,

the former vehicle is waiting for the end of loading and then is being

loaded itself. Then the vehicles movement is modeled by a dynamical system of the

class to this system.

\vskip 3pt

{P. 11--12.}

\vskip 1pt

\section*{References}

{1] von Neumann J. 1963. The general and logical theory of automata.

In {\it J. von Neumann, Collected works}, edited by A.H. Taub, 5, 288.

\vskip 3pt

[2] von Neumann J. 1966. {\it Theory of self-reproducing automata,

edited by A.W. Burks} (University of Illions, Urnana).

\vskip 3pt

[3] Ulam S. 1974. {\it Some ideas and prospects in bio-mathematics.} Ann. Rev. Bio. 255.

\vskip 3pt

[4] Wolfram, S. {\it Statistical mechanics of cellular automata.}

Rev. Mod. Mod. Phys. 1983, {\it Vol.~55,} pp.~601--644.\\

https:/dx.doi.org.10.1003/RevModPhys.55.601

\vskip 3pt

[5] Wolfram S. Tables of cellular automaton properties.

In {\it Theory and Applications of Cellular Automata (Including Selected

papers 1983--1986)}; [Wolfram S. (Ed.)] Advanced Series on Complex

Systems 1. World Scientific Publishing, 1986, pp.~485--557.

\vskip 3pt

[6] Buslaev A.P., Tatashev A.G., Yashina M.V. {\it On cellular

automata, traffic and dynamical systems in graphs.} International

Journal of Engineering and Technology,  vol.~7, no.~2.28, pp.~351--356.\\

https://dx.doi.org/10.14419/ijet.v7i2.28.13210

\vskip 3pt

[7] Kozlov V.V., Buslaev A.P., Tatashev A.G.

{\it Monotonic walks on a necklace and coloured

dynamic vector.} International Journal of Computer

Mathematics, vol. 92, no. 9, 2015, pp. 1910~-- 1920.

1920.\\

https://dx.doi.org/1080/00207160.2014/915964

\vskip 3pt

[8] Buslaev  A.P., Tatashev  A.G., Fomina M.J.,  Yashina M.V.  On Spectra

of Wolfram Cellular Automata in Hamming Spaces. In Proceedings of the

6th International Conference on Control, Mechatronics and Automation Tokyo,

Japan — October 12 - 14, 2018, ACM New York, NY, USA, 2018. pp. 123-127.\\

doi: 10.1145/3284516.3284549

[9] Nagel K., Schreckenberg M. {\it A cellular automation models

for freeway traffic.} J. Phys. I. France 2, {\bf 1992}, pp. 2221~-- 2229.\\

https://dx.doi.org/10.1051/jp1.1992277

[10] Belitzky V., Ferrary P.A. {\it Invariant measures and convergence

properties for cellular automation 184 and related processes.}

J. Stat. Phys. (2005), vol.~118, no.~3, pp.~589--623\\

https://dx.doi.org/10.1007/s10955-044-8822-4

[11] Blank M. {\it Exact analysis of dynamical systems arising

in models of flow traffic.} Russian Math Surveys, 2000, vol.~55, no.~55,~pp.~562--563

(DOI: 10.4213/rm295) {\bf 55(5)} 562-563

[12] Gray L and Grefeath D

The ergodic theory of traffic jams. {\it J. Stat. Phys.},

(DOI: 10.1023/A:1012202706850) {\bf 105(3/4)} 413--452

[13] Kanai M, Nishinary K and Tokihiro T. Exact solution and

asymptotic behavior of the asymmetric simple exclusion

process on a ring {\it arXive.0905.2795v1 [cond-mat-stat-mech] 18 May 2009}

[14] Biham O, Middleton A A and Levine D 1992

Self-organization and a dynamical transition in traffic-flow models

{\it Phys. Rev. A. American Physical Society} (DOI 10.1103/PhysRevA.46.R6124)

{\bf 46(10)} R6124--R6127

[15] D'Souza R M (2005) Coexisting phases and lattice dependence

of a cellular automaton model for traffic flow  {\it Phys. Rev. E.

The American Physical Society} (DOI 10.1103/PhysRevA46.R6.124) {\bf 71(6)}: 066112

[16] Angel O, Horloyd A E, Martin J B. (12 August 2005).

The jammed phase of the Biham-Middleton-Levine traffic model

{\it Electronic Communication in Probability} (DOI 10.1214/ECP.v10-1148)

{\bf 10}: 167--178

[17] Malecky K. {\it Graph cellular automata with relation-based neighbourhoods

of cells for complex systems modelling: A case of traffic simulation.} Symmetry, 2017, 9,  p.~322.

https://doi.org/103390/sym912032

[18] Buslaev A.P., Tatashev A.G., Yashina M.V. About synergy of flows on flower. In Dependability

Engineering and Complex Systems. Springer, Cham {\bf 2016.} pp. 75--84.

[19] Buslaev A.P., Tatashev A.G. {\it Exact Results for discrete dynamical

system on a pair of contours.} Mathematical Methods in the

Applied Sciences, 2019, vol.~41, no.~17, pp.~7283--7204.\\

https://dx.doi.org/10.1002/mma.4822.

[20] Buslaev A.P., Fomina M.Yu., Tatashev A.G., Yashina M.V.

{\it On discrete flow networks model spectra:

statement, simulation, hypotheses.} Journal of Physics:

Conference Series, (2018), vol.~1053, 012034, pp.~1--7\\

https://dx.doi.org.10.1088/1742/6596/1053/1/012034

[21] Buslaev A.P., Tatashev A.G., Yashina M.V.

{\it Flows spectrum on closed trio of contours.} European

Journal of Pure and Applied Mathematics, 2018, vol. 11, no. 1,

pp. 260--283.\\

http://dx.doi.org.10.29020/nybg.ejpam.v11i1.3201

[22] Zubilage D., Cruz G., Aguilar L.D., Zapotecatl J., Fernandes N.,

Aguilar J., Rosenblueth D.A., Gershenson C. {\it Measuring the Complexity

of Self-Organizing Traffic Lights.} Entropy, 2014, vol.~16, no.~5, pp.~2384-2407.\\

https://doi.org/10.3390/e16052384

[23] Kozlov V.V., Buslaev A.P., Tatashev A.G. {\it A dynamical communication system

on a network.} Journal of Computational and Applied Mathematics, 2015, vol.~275, pp.~247--261.\\

https://doi.org/10.1016/j.cam.2014.07.026

Reviewer 3 Report

This study investigated the spectrum of some elementary cellular automata. A dynamical system was introduced which is a version of the binary chain of contours and called a generalization of the binary closed chain of contours. Moreover, the concept of the spectrum for a class of cellular automata was presented with a claim that this definition is more general than the classification provided by Wolfram for elementary cellular automata. Then, the authors found that generalized closed chain belongs to a class of cellular automata.

Although the topic is interesting, the structure of paper as well as several ambiguous in presenting and proving ideas convince the reviewer to reject it.

I can conclude my concerns and comments as follows:

¾     P.8, L.203: When two different definition goes for one phenomenon (eigenvalue), the soundness of all theories is in doubt. Here, you defined the eigenvalue based on a limitation while in P. 7, L. 157, the definition is based on a summation. How can you relate these definitions?

¾     Although the abstract is informative, I suggest to re-write it more smoothly. For instance, authors can use “Contributions towards this study are…” instead of several short sentences start with “We”

¾     P.1, L. 15: determined -> defined.P.1, L.18: S. Ulam have introduced -> S. Ulam introduced. P.1, L. 21: cellular automaton has been introduced -> cellular automaton was introduced, etc. There is a lot of such typos and odd usage of English in the paper which must be resolved.

¾     P.1, L. 23-26: the sentences do not make sense from a grammatical perspective.

¾     P.1, L. 30: spectrum of generalized elementary cellular automata -> Generalized CA? What does it mean? We have 256 rule, and it was proved that rule 110 could be a universal Turing machine. However, I cannot understand this.

¾     P.1, L.32: CA029 -> Do you mean ECA rule 29 with this notation? This is your notation, so it is better first to define it then use it. You postponed the definition to Section 2. Moreover, when you define your notation in Section 2, there is a contradiction. This notation is not also compatible with that of already used. Because rule 1 is expected to be written as CA 001 in this notation.

¾     P,2, L.36: It is evident that $H^{n} = {x}$ is the unit cube in $R^{n}$, $card(H^{n}) = 2^{n}$ -> It is not evident. Because this definition is not mathematically sound.

1) Sometimes the term "unit cube" refers in specific to the set $[0, 1]^{n}$ of all n-tuples of numbers in the interval [0, 1].

2) An n-tuple is a sequence of n elements, where n is a non-negative integer. An n-tuple usually writes in the form of listing the elements within parentheses "()" and separated by commas; for example, (2, 7, 4, 1, 7)

¾     $H^{n}$: In standard math, this notation is used to refer to Hilbert space! So, it’s recommended to use the mathematical notation with caution.

¾     P.2, L38: Hamming space -> a Hamming space can be defined over any alphabet as the set of words of a fixed length N with letters from Q. However, you assume that you work on an infinite sequence which is in contradiction to this notation! Moreover, the base distance in Hamming space is the Hamming distance, not city-block distance as you defined here. If you decide to use Hamming distance, you must prove the complexity of its calculation when the input string of ECA is large enough!

¾     P.2, Section 3: the cellular automaton states are repeated periodically from some moment -> It is wrong because some rules show chaos!

¾     P.3: Denoted by $T = T(x(0))$ … The whole sentence must paraphrase.

¾      with (1); then the ergodic limit equals -> To prove this equivalent, you must define the following. Let U be a unitary operator on a Hilbert space H; more generally, an isometric linear operator (that is, a not necessarily surjective linear operator) satisfying ‖Ux‖ = ‖x‖ for all x in H, or equivalently, satisfying U*U = I, but not necessarily UU* = I. Therefore, you must prove that there is a P which is the orthogonal projection onto {ψ H | Uψ = ψ} = ker(I − U). However, these considerations and proof were not provided by the authors.

¾     P.4, L.78: Let us denote by [a] the integer part of [a] -> Does this notation show integral part of [a] or a? A lot of typos in mathematical formulations are not acceptable.

¾     The spectrum of eigenvalues for cellular automata -> Refer to the following equation, which unfortunately does not have a number for referring to, you mean the eigenvalue spectrum is the same as velocities? You must provide proof of that because it is not trivial.

¾     Proof of Theorem 1: This proof is full of contradiction, because in the previous sentence authors mentioned x_{i-1}(t) = 0 and x_{i+1}(t) = 1. Then how is it possible that x_{i-1}(t) = x_{i+1}(t)?

¾     For the rest of theorems and proofs: Both writing and formulation of theorems have many difficulties for presentation. The proofs are not strong enough to follow the theory itself. Moreover, proofs of some theories are based on the previous one which in this way we can easily see nested wrongness.

¾     P.7, L.187: There is a question for the reviewer to this point that what the exact definition of an eigenvalue in ECA is? The boldest shortcoming of this paper is lack of proper structure and a followable storyline. Although the idea is interesting, tracking the paper for the audience is hard. So, my main comment is a general re-structuring of paper to reach a better and easy to follow manuscript. Moreover, after restructuring, authors must ask a native speaker to have a review on the paper because the number of typos and odd usage of English is higher than a manuscript submitted to a journal.

Author Response

Thank you so much for deep  comments.

We have made corrections in the text of the paper and, in particular, in the text of proofs. We have extended the abstracts, introduction and the list of references.

We have made the following corrections.

{\bf P.8, L.202. P.9, L. 203.}

\vskip 3pt

The average value of the $\rho(x,y)$ is called {\it an eigenvalue.}

\vskip 3pt

{\bf P.2.} Suppose $\Delta x(t)=\rho(x(t),y(t)),$ where $\rho$ is defined in

accordance with (1); then the value

$$\frac{1}{T}\sum\limits_{t=t_0}^{T-1}\Delta x(t)=\lambda(x(0))$$

is called the {\it eigenvalue} of the spectral cycle.

\vskip 3pt

{\bf P.1, Abstracts.} In this paper, we study the properties of some elementary automata.

We have obtained characteristics of these cellular automata.

The concept of the spectrum for more general class than

the class of elementary automata has been introduced. We introduce and study discrete dynamical

systems which are mass transport on closed chains of contours. Particles on contours

in accordance with given rules. These dynamical systems can be interpreted as

cellular automata. Contributions towards this study are

the following. Characteristics of some elementary cellular

automata have been obtained. A theorem about the velocity

of particles movement on the closed chain has been proved.

It has been proved that, for any $\varepsilon>0,$

there exists a chain with flow density $\rho<\varepsilon$ such

that the average flow particle velocity is less than the velocity

of free movement. An interpretation of this system as a transport

model is given. The spectrum of a binary closed chain with some

conflict resolution rule is studied.

\vskip 3pt

{\bf P.1, L.15.} L. 18., L. 21., L.~23--26. We have made corrections.

\vskip 3pt

P.2, L.29--30. In this paper, we continue to study the spectrum

of elementary automata and introduce the spectrum of a class  of cellular automata.

\vskip 3pt

{\bf P.1, L.27.} It has been noted in [8] that this system is the elementary cellular

automaton CA~063 (or ECA rule 063).

\vskip 3pt

{\bf P. 2. L.34}

There are 256 elementary cellular automata $CA~000,$ $CA~001,\dots,CA~255.$

\vskip 3pt

{\bf P.2, L.34--L.38.}

\vskip 3pt

We consider a set of $n$-dimensional vectors

$x=(x_1,\dots,x_n)$ with binary coordinates $x_i\in \{0,1\}.$

We introduce the distance function

$$\rho(x,y)=\frac{1}{n}\sum\limits_{i=1}^n|x_i-y_i|.\eqno(1)$$

\vskip 3pt

{\bf P.2, L. 39--40.} Let us denote by $X(x(0))$ the spectral cycle. Let us denote by

$T=T(x(0))$ the period of the spectral cycle.

\vskip 3pt

{\bf P.2, L.39--40.} Suppose a cellular automaton is defined on a closed circle.  Since the cellular automaton...

\vskip 3pt

{\bf P.4, L.78--79.} Let us denote by $[a]$ the integral part of $a.$

\vskip 3pt

{\bf P.4.} {\bf Theorem 1.} {\it Suppose the number of a binary closed chain contours is $n,$

and the initial state is a vector

$$(\alpha_1(t),\dots,\alpha_n(t)),$$

where, if the particle of the $i$th contour is in the lower cell at time $t,$ then

$\alpha_i(t)=0,$ and, if the particle of the $i$th contour is in the upper cell,

then $\alpha_i(t)=1.$  Particles move counter-clockwise, and the conflict resolution

rule is the right-priority rule. Then the average velocity of particles is equal to

the eigenvalue of the relative spectral cycle of CA~063 that is defined on the ring,

containing $i$ particles.

\vskip 3pt

Proof.} Suppose the initial states of the closed chain and the cellular automaton

are $(\alpha_1(0),\dots,\alpha_n(0)),$ $(x_1(0),\dots,x_n(0))$ respectively,

and $\alpha_i(0)=x_i(0).$ Then we have $\alpha_i(t)=x_i(t),$ $i=1,\dots,n$ for any

$t=0,1,2,\dots$ The coordinate $x_i(t)$ is changed at time $t$ if and only if the

particle of the $i$th contour moves at time $t.$ From this, the theorem follows.

\vskip 3pt

{\bf P.5. L.67--69.} If at time $t+1$ the cell is in the state 1 $x_i(t+1)=1,$ then $x_{i-1}=x_i(t)=0,$

$x_{i+1}=1$ (indexes are calculated by modulo $n).$ Therefore, in accordance

with rule CA 002, $x_{i-2}(t+1)=x_{i-1}(t+1)=x_{i+1}(t+1)=x_{i+2}(t+1).$

\vskip 3pt

{\bf P. 8. L.176.} ''0'' in the case of CA~255

\vskip 3pt

{\bf P.7. L.148.} From this, Theorem~10 follows in the case of rule $CA~010.$ In the case of CA~014, CA~024

\vskip 3pt

{\bf P.2. Section 1.} In this paper, we continue to study the spectrum

of elementary automata and introduce the spectrum of a class  of cellular automata. We

study a generalization of the binary closed chain of contours. This system belongs to

the class of generalized elementary automata.

We also study a version of a binary closed chain

of contours, which is the elementary cellular automaton CA~029.

In Sections 2, 3, the concept of the spectrum of an elementary cellular

automaton is introduced.

In Section 4, we give a concise overview on traffic models based on cellular

automata.

In Section 5, we describe a dynamical system called a binary chain of

contours. This system is called a closed chain of contours with the left--priority

conflict resolution rule. This system is equivalent to CA~063. The concept of the average

velocity of particles is introduced. We have proved that the value of the average

velocity and the eigenvalue of CA~063 are the same.

In Section~6, we prove theorems about properties of some elementary

automata. In particular, we study the spectrum of elementary cellular

automata.

In Section 7, we study properties of a version of binary closed chain of contours.

This system is equivalent to CA~029. A theorem about the spectrum of the system

has been proved.

In Sections 8, 9, we introduce the concept of the spectrum is defined for a class of cellular

automata more general than the class of elementary cellular automata.

This system is interpreted as a cellular automaton. We prove a theorem

about properties of the system. Let us describe a transport interpretation

of this  system. Suppose raw materials or fuel is delivered to

subdivisions of a factory from warehouses by vehicles, for example

trucks moving on factory railways. Each vehicle moves on its line.

The lines cross at points such that warehouses are located in these points.

If one of vehicles comes to a warehouse while another vehicle is being loaded,

the former vehicle is waiting for the end of loading and then is being

loaded itself. Then the vehicles movement is modeled by a dynamical system of the

class to this system.

\vskip 3pt

{P. 11--12.}

\vskip 1pt

\section*{References}

{1] von Neumann J. 1963. The general and logical theory of automata.

In {\it J. von Neumann, Collected works}, edited by A.H. Taub, 5, 288.

\vskip 3pt

[2] von Neumann J. 1966. {\it Theory of self-reproducing automata,

edited by A.W. Burks} (University of Illions, Urnana).

\vskip 3pt

[3] Ulam S. 1974. {\it Some ideas and prospects in bio-mathematics.} Ann. Rev. Bio. 255.

\vskip 3pt

[4] Wolfram, S. {\it Statistical mechanics of cellular automata.}

Rev. Mod. Mod. Phys. 1983, {\it Vol.~55,} pp.~601--644.\\

https:/dx.doi.org.10.1003/RevModPhys.55.601

\vskip 3pt

[5] Wolfram S. Tables of cellular automaton properties.

In {\it Theory and Applications of Cellular Automata (Including Selected

papers 1983--1986)}; [Wolfram S. (Ed.)] Advanced Series on Complex

Systems 1. World Scientific Publishing, 1986, pp.~485--557.

\vskip 3pt

[6] Buslaev A.P., Tatashev A.G., Yashina M.V. {\it On cellular

automata, traffic and dynamical systems in graphs.} International

Journal of Engineering and Technology,  vol.~7, no.~2.28, pp.~351--356.\\

https://dx.doi.org/10.14419/ijet.v7i2.28.13210

\vskip 3pt

[7] Kozlov V.V., Buslaev A.P., Tatashev A.G.

{\it Monotonic walks on a necklace and coloured

dynamic vector.} International Journal of Computer

Mathematics, vol. 92, no. 9, 2015, pp. 1910~-- 1920.

1920.\\

https://dx.doi.org/1080/00207160.2014/915964

\vskip 3pt

[8] Buslaev  A.P., Tatashev  A.G., Fomina M.J.,  Yashina M.V.  On Spectra

of Wolfram Cellular Automata in Hamming Spaces. In Proceedings of the

6th International Conference on Control, Mechatronics and Automation Tokyo,

Japan — October 12 - 14, 2018, ACM New York, NY, USA, 2018. pp. 123-127.\\

doi: 10.1145/3284516.3284549

[9] Nagel K., Schreckenberg M. {\it A cellular automation models

for freeway traffic.} J. Phys. I. France 2, {\bf 1992}, pp. 2221~-- 2229.\\

https://dx.doi.org/10.1051/jp1.1992277

[10] Belitzky V., Ferrary P.A. {\it Invariant measures and convergence

properties for cellular automation 184 and related processes.}

J. Stat. Phys. (2005), vol.~118, no.~3, pp.~589--623\\

https://dx.doi.org/10.1007/s10955-044-8822-4

[11] Blank M. {\it Exact analysis of dynamical systems arising

in models of flow traffic.} Russian Math Surveys, 2000, vol.~55, no.~55,~pp.~562--563

(DOI: 10.4213/rm295) {\bf 55(5)} 562-563

[12] Gray L and Grefeath D

The ergodic theory of traffic jams. {\it J. Stat. Phys.},

(DOI: 10.1023/A:1012202706850) {\bf 105(3/4)} 413--452

[13] Kanai M, Nishinary K and Tokihiro T. Exact solution and

asymptotic behavior of the asymmetric simple exclusion

process on a ring {\it arXive.0905.2795v1 [cond-mat-stat-mech] 18 May 2009}

[14] Biham O, Middleton A A and Levine D 1992

Self-organization and a dynamical transition in traffic-flow models

{\it Phys. Rev. A. American Physical Society} (DOI 10.1103/PhysRevA.46.R6124)

{\bf 46(10)} R6124--R6127

[15] D'Souza R M (2005) Coexisting phases and lattice dependence

of a cellular automaton model for traffic flow  {\it Phys. Rev. E.

The American Physical Society} (DOI 10.1103/PhysRevA46.R6.124) {\bf 71(6)}: 066112

[16] Angel O, Horloyd A E, Martin J B. (12 August 2005).

The jammed phase of the Biham-Middleton-Levine traffic model

{\it Electronic Communication in Probability} (DOI 10.1214/ECP.v10-1148)

{\bf 10}: 167--178

[17] Malecky K. {\it Graph cellular automata with relation-based neighbourhoods

of cells for complex systems modelling: A case of traffic simulation.} Symmetry, 2017, 9,  p.~322.

https://doi.org/103390/sym912032

[18] Buslaev A.P., Tatashev A.G., Yashina M.V. About synergy of flows on flower. In Dependability

Engineering and Complex Systems. Springer, Cham {\bf 2016.} pp. 75--84.

[19] Buslaev A.P., Tatashev A.G. {\it Exact Results for discrete dynamical

system on a pair of contours.} Mathematical Methods in the

Applied Sciences, 2019, vol.~41, no.~17, pp.~7283--7204.\\

https://dx.doi.org/10.1002/mma.4822.

[20] Buslaev A.P., Fomina M.Yu., Tatashev A.G., Yashina M.V.

{\it On discrete flow networks model spectra:

statement, simulation, hypotheses.} Journal of Physics:

Conference Series, (2018), vol.~1053, 012034, pp.~1--7\\

https://dx.doi.org.10.1088/1742/6596/1053/1/012034

[21] Buslaev A.P., Tatashev A.G., Yashina M.V.

{\it Flows spectrum on closed trio of contours.} European

Journal of Pure and Applied Mathematics, 2018, vol. 11, no. 1,

pp. 260--283.\\

http://dx.doi.org.10.29020/nybg.ejpam.v11i1.3201

[22] Zubilage D., Cruz G., Aguilar L.D., Zapotecatl J., Fernandes N.,

Aguilar J., Rosenblueth D.A., Gershenson C. {\it Measuring the Complexity

of Self-Organizing Traffic Lights.} Entropy, 2014, vol.~16, no.~5, pp.~2384-2407.\\

https://doi.org/10.3390/e16052384

[23] Kozlov V.V., Buslaev A.P., Tatashev A.G. {\it A dynamical communication system

on a network.} Journal of Computational and Applied Mathematics, 2015, vol.~275, pp.~247--261.\\

https://doi.org/10.1016/j.cam.2014.07.026

Round 2

Reviewer 3 Report

Authors answered the reviewer's concerns. However, this paper still suffers from the method of writing. It is highly recommended to ask a native speaker to paraphrase sentences, especially in the abstract and introduction parts to make the paper readable. 

Author Response

We have revised our manuscript.
